# Involvement of NINJ2 Protein in Inflammation and Blood–Brain Barrier Transmigration of Monocytes in Multiple Sclerosis

**DOI:** 10.3390/genes13111946

**Published:** 2022-10-25

**Authors:** Melissa Sorosina, Silvia Peroni, Elisabetta Mascia, Silvia Santoro, Ana Maria Osiceanu, Laura Ferrè, Ferdinando Clarelli, Antonino Giordano, Miryam Cannizzaro, Filippo Martinelli Boneschi, Massimo Filippi, Federica Esposito

**Affiliations:** 1Laboratory of Neurological Complex Disorders, Division of Neuroscience, Institute of Experimental Neurology (INSPE), IRCCS San Raffaele Scientific Institute, 20132 Milan, Italy; 2Neurology and Neurorehabilitation Unit, IRCCS San Raffaele Scientific Institute, 20132 Milan, Italy; 3Neurology Unit, IRCCS Fondazione Ca’ Granda Ospedale Maggiore Policlinico, 20122 Milan, Italy; 4Department of Health Sciences, University of Milan, 20122 Milan, Italy; 5Vita-Salute San Raffaele University, 20132 Milan, Italy; 6Neurophysiology Unit, IRCCS San Raffaele Scientific Institute, 20132 Milan, Italy; 7Neuroimaging Research Unit, Division of Neuroscience, Institute of Experimental Neurology (INSPE), IRCCS San Raffaele Scientific Institute, 20132 Milan, Italy

**Keywords:** multiple sclerosis, inflammation, transmigration

## Abstract

Multiple sclerosis (MS) is an inflammatory neurodegenerative disorder of the central nervous system (CNS). The migration of immune cells into the CNS is essential for its development, and plasma membrane molecules play an important role in triggering and maintaining the inflammation. We previously identified ninjurin2, a plasma membrane protein encoded by *NINJ2* gene, as involved in the occurrence of relapse under Interferon-β treatment in MS patients. The aim of the present study was to investigate the involvement of NINJ2 in inflammatory conditions and in the migration of monocytes through the blood–brain barrier (BBB). We observed that NINJ2 is downregulated in monocytes and in THP-1 cells after stimulation with the pro-inflammatory cytokine LPS, while in hCMEC/D3 cells, which represent a surrogate of the BBB, LPS stimulation increases its expression. We set up a transmigration assay using an hCMEC/D3 transwell-based model, finding a higher transmigration rate of monocytes from MS subjects compared to healthy controls (HCs) in the case of an activated hCMEC/D3 monolayer. Moreover, a positive correlation between *NINJ2* expression in monocytes and monocyte migration rate was observed. Overall, our results suggest that ninjurin2 could be involved in the transmigration of immune cells into the CNS in pro-inflammatory conditions. Further experiments are needed to elucidate the exact molecular mechanisms.

## 1. Introduction

Multiple sclerosis (MS) is an inflammatory disease of the central nervous system (CNS) characterized by multifocal areas of leukocyte infiltration, demyelination and axonal degeneration [1]. It is a multifactorial disorder in which autoreactive T cells are known to migrate, along with other immune cells, from the peripheral blood inside the CNS through the blood–brain barrier (BBB). Macrophages dominate the infiltrate, followed by CD4^+^ and CD8^+^ T lymphocytes and, in lower numbers, by B lymphocytes and plasma cells [2]. Secretion of pro-inflammatory factors inside the CNS leads to an inflammatory amplification causing myelin and axonal damage [3]. In this context, plasma membrane molecules seem to be essential in triggering and maintaining the inflammation in MS. This is also supported by the fact that some approved effective drugs, such as natalizumab and fingolimod, are able to target selectively specific membrane proteins, blocking or reducing the ability of immune cells to reach the CNS [4,5].

We previously identified a specific membrane protein, ninjurin2, encoded by the *NINJ2* gene, as implicated in the response to Interferon-β (IFN-β) in MS patients. Specifically, through a pharmacogenetic approach, we identified an association between rs7298096, located in an enhancer region ~3.3 kb upstream of the *NINJ2* gene and associated with its expression [6], and the long-term response to IFN-β treatment in MS patients (rs7298096_A_: OR = 1.96, *p* = 9.81 × 10^−5^) [7]. The analyses further revealed that this association is mainly related to the time of occurrence of clinical inflammatory activity (time to first relapse, rs7298096_AA_: hazard ratio = 1.41, *p* = 3 × 10^−4^) [6], suggesting that this SNP and *NINJ2* are involved in the inflammatory component of the disease. This association was confirmed in a more recent study demonstrating that *NINJ2* gene expression is higher in non-responders compared to responder IFN-β-treated MS patients [8].

The ninjurin2 protein is an adhesion molecule involved in neurite outgrowth and neuronal regeneration through homophilic interactions [9]. In addition to cells of the nervous system, NINJ2 is also highly expressed in peripheral leukocytes, lymph nodes, bone marrow and lungs [9], as well as in CD14^+^ monocytes and CD33^+^ myeloid cells [7]. This gene has been associated with several neurological disorders, including stroke [10,11,12,13,14,15] and Alzheimer’s disease [16], but it also seems implicated in other diseases, including obsessive–compulsive disorder [17], neuropathic pain [18], diabetes mellitus type 1 [19] and cancer [20,21]. The involvement of NINJ2 in MS is suggested by recent reports, although its exact role in the disease has not yet been elucidated. Specifically, a “suggestive” genetic association of *NINJ2* variants with susceptibility to MS has been observed [22], as well as a different methylation state in CD4^+^ and CD8^+^ lymphocytes, coupled with a different *NINJ2* expression, when comparing MS patients with HCs [23]. In the context of inflammation, NINJ2 was found to act as a pro-inflammatory mediator, having a role in the inflammation processes and in endothelial cell activation through its interaction with the TLR4/NF-kB pathway, as well as in the adhesion of monocytes to endothelial cells [24]. Our hypothesis is that ninjurin2 could also be implicated in the transmigration of immune cells during inflammatory processes in MS, affecting in turn the disease activity and explaining the observed relationship between *NINJ2* genetic variants, gene expression and MS relapses, as suggested by previous pharmacogenetic studies. The aim of the present study is to understand whether *NINJ2* expression is regulated in inflammatory conditions and its involvement in monocyte migration through the BBB.

## 2. Materials and Methods

### 2.1. Pro-Inflammatory Stimulation Assays in Monocytes

Stimulation assays were performed using samples collected from 7 healthy controls (HCs; 5 females, 2 males; mean age (years) ± standard deviation (SD) = 30.1 ± 3.2). Whole blood was collected in Vacutainer^®^ Cellular Preparation Tubes (BD) and Peripheral Blood Mononuclear Cells (PBMCs) were isolated according to the manufacturer’s instructions. CD14^+^ monocytes were then isolated using CD14 Microbeads (Miltenyi Biotec) following the manufacturer’s protocol and cultured in RPMI1640 medium. After isolation, CD14^+^ monocytes (10^6^ cells/wells) were incubated for 6 h or 24 h, either unstimulated or stimulated with TNFα (10 ng/mL), IFNγ (100 ng/μL) or LPS (1 μg/mL). After stimulation, total RNA was extracted using TRIzol^TM^ reagent (Invitrogen^TM^) according to manufacturer’s instructions; quantity and integrity of RNA were assessed using a Nanodrop-8000 spectrophotometer (Thermo Fisher, Waltham, MA, USA) and RNA 6000 Nano Kit on an Agilent 2100 Bioanalyzer. RNA was reverse transcribed with the High-Capacity RNA-to-cDNA Retrotranscription kit (Applied Biosystems, Waltham, MA, USA) and *NINJ2* gene expression was measured through qRT-PCR on ViiA7 System (Thermo Fisher) using pre-designed TaqMan probes (Thermo Fisher assay code: Hs04399286_m1) according to the standard protocol for TaqMan assays (fast modality). The relative expression, taking an arbitrary reference sample as 1, was calculated from technical replicates normalized to ACTB gene (Thermo Fisher assay code: Hs01060665_g1) with the 2^−ΔΔCt^ method.

Changes at protein level of NINJ2 upon incubation with pro-inflammatory stimuli were also evaluated in 3 HCs (3 females; mean age (years) ± SD = 32.4 ± 7.6) through immunocytochemistry (ICC). Monocytes unstimulated or stimulated with LPS (1 μg/mL) for 24 h were spotted as monolayers on slides using Cytospin^TM^ Centrifuge. Cells were fixed with 4% paraformaldehyde solution, non-specific binding sites were blocked with 5% horse serum/0.1% Triton X, and monocytes were labelled with anti-NINJ2 (rabbit polyclonal antibody, Novus Biologicals, code: NBP2-56698) 1:50 and anti-CD68 (Agilent Technology) 1:100 primary antibodies and with anti-rabbit Alexa 546 and anti-mouse Alexa 488 secondary antibodies (Invitrogen^TM^). Slides were then labelled with 1:1000 Hoechst 33,342 (Sigma-Aldrich) and mounted with Fluoromount-G™ Mounting Medium (Invitrogen^TM^). Stained coverslips were imaged using a Leica TCS SP8 confocal microscope with a 63× objective. For each sample, NINJ2 protein expression was measured starting from the generated images according to ImageJ software v1.53 [25], quantifying it in at least 5 cells. Due the high variability among the different samples, the relative protein expression was measured choosing an arbitrary reference cell as 1 for each individual.

### 2.2. Pro-Inflammatory Stimulation Assays in THP-1 Cells

The leukemic monocyte cell line (THP-1) was bought from Sigma-Aldrich and cultured with RPMI1640 medium supplemented with 2 mM glutamine and 10% Fetal Bovine Serum (FBS). The effect of LPS stimulation on THP-1 cell line was evaluated as was done for monocytes, adding LPS (1 μg/mL) (stimulated condition) or PBS (unstimulated control condition) to the media for up to 24 h. Next, qRT-PCR was used to measure the relative expression of *NINJ2* and *IL6* expression (Thermo Fisher assay code: Hs00985639_m1), as described. ICC was performed as described for monocytes. Each experiment was performed in duplicate, with NINJ2 protein expression being measured starting from the generated images according to ImageJ software v1.53 [25], quantifying it in at least 10 cells for each condition.

### 2.3. Pro-Inflammatory Stimulation Assays in hCMEC/D3

The human brain microvascular endothelial cell line (hCMEC/D3) was bought from Merck and cultured with EndoGRO^TM^-MV Complete Media Kit (Merck) supplemented with 1 ng/mL FGF-2 (Merck) in collagen-coated support. The effect of LPS stimulation on hCMEC/D3 cell line was evaluated as performed for monocytes, adding LPS (1 μg/mL) (stimulated condition) or PBS (unstimulated control condition) to the media for up to 24 h, followed by qRT-PCR as described for monocytes using GADPH (Thermo Fisher assay code: Hs02758991_g1) as endogenous control. ICC was performed as described previously. Each experiment was performed in duplicate, analyzing a total of 12 cells for each condition.

### 2.4. Transmigration Assay

A total of 7 untreated MS patients were recruited (female:male ratio = 4:3, mean age (years) ± SD = 51.9 ± 8.6; clinical course: 1 relapsing–remitting MS, 5 primary progressive MS, 1 secondary progressive MS), as well as 7 age- and sex-matched HCs (female:male ratio = 4:3, mean age ± SD = 56.9 ± 6.3, *t*-test *p*-value HC vs. MS = 0.25). Patients were diagnosed as MS according to the McDonald 2017 criteria [26]. Monocytes were collected as described for the pro-inflammatory stimulation assays.

Transmigration assay was performed using hCMEC/D3 transwell-based model according to the CytoSelect^TM^ Leukocyte Transmigration Assay kit (Cell Biolabs). hCMEC/D3 are widely used as a model of human BBB [27] and were chosen to create a surrogate of the BBB. Briefly, 100,000 cells/well hCMEC/D3 were seeded on collagen type 1-coated porous (3 μm) polycarbonate membrane inserts in 24 transwell plates and cultured in EndoGRO^TM^-MV Complete Media Kit (Merck) supplemented with 1 ng/mL FGF-2 (Merck). After 24 h, the EndoGRO^TM^-MV Complete Media Kit was changed with EndoGRO^TM^-MV Complete Media Kit containing LPS (1 μg/mL) or an equal volume of PBS (unstimulated control) and the entire plate was then incubated at 37 °C with 5% CO_2_ for an additional 24 h. Then, 100,000 THP-1 cells or freshly collected monocytes in 100 μL of RPMI1640 growth media previously labelled with florescent LeukoTracker^TM^ dye were added (with or without LPS (1 μg/mL)) and this insert was transferred to a new 24 transwell plate filled with 500 μL RPMI1640 media supplemented with FBS (10%). Cells were allowed to migrate for 24 h. Migrated cells were then quantified after lysis on a Victor^3^ fluorescence plate reader (PerkinElmer) at 480 nm/520 nm in triplicate and the mean value was used for the statistical analyses. Transmigration experiments were performed in 3 conditions: (1) without stimulation, (2) pre-stimulating the hCMEC/D3 cells’ monolayer with LPS (1 μg/mL) for 24 h, (3) pre-stimulating hCMEC/D3 cells’ monolayer with LPS (1 μg/mL) for 24 h and stimulating monocytes/THP-1 with LPS (1 μg/mL) during the migration (24 h). For fresh monocytes, each migration assay was performed pairing one MS with one matched HC and each condition was performed in duplicate, while for THP-1, the experiment was performed in triplicate. Barrier integrity was verified for each experiment quantifying the amount of 150 kDa FITC-dextran (Sigma-Aldrich) in the lower chamber 30 min after its addition (100 μg) to the upper chamber by fluorimetric analysis (Victor^3^ plate reader, PerkinElmer).

In order to measure the basal expression of *NINJ2* in collected monocytes and to correlate it with the migration rate, for each individual, total RNA was extracted from 10^6^ monocytes and qRT-PCR was performed as described.

### 2.5. Statistical Analyses

Statistical analyses were performed using R (version 4.1.3) and Prism version 8.2.1 (GraphPad Software, La Jolla, CA, USA). Comparisons between groups were performed according to a two-tailed parametric (Student’s *t*-test) or non-parametric (Mann–Whitney test) test according to departure from normality of the distribution of values, unpaired or paired based on the design of the experiment. Two-way repeated measures ANOVA was used to compare HC and MS migration rate during the transmigration assay across the different stimulation settings. Deming regression was used to assess the relationship between migration rate and expression of *NINJ2*. Statistical significance was set at *p* < 0.05.

## 3. Results

### 3.1. NINJ2 is Down-Regulated after Pro-Inflammatory Stimulation in Monocytes and in THP-1 Cells

We measured the expression of *NINJ2* gene in the presence of three pro-inflammatory cytokines: LPS, IFNγ and TNFα. In monocytes, we observed a downregulation of *NINJ2* under LPS stimulation, which was more evident after short-term stimulation (6 h, *p* = 0.0074, paired *t*-test) compared to 24 h (*p* > 0.05, paired *t*-test), as shown in Figure 1A. No major changes were observed when stimulating cells with IFNγ (Figure 1B) or TNFα (Figure 1C). The effect of LPS was also confirmed at the protein level, as shown in Figure 1D, suggesting that pro-inflammatory stimulation with LPS leads to a downregulation of NINJ2 in monocytes.

We also measured the expression of *NINJ2* gene in presence of LPS in THP-1 cells, a cell line derived from human monocytes isolated from a patient with acute monocytic leukemia. Similarly to what we noticed for monocytes, we observed a downregulation of *NINJ2* under LPS stimulation (Figure 1E,F). *IL6* gene expression was used to evaluate the activation state of cells. As expected, our results confirm that stimulation with LPS leads to cell activation and that there is a negative correlation between cell activation and *NINJ2* expression (Pearson r = − 0.84; *p* = 7 × 10^−4^).

### 3.2. NINJ2 is Up-Regulated after LPS Stimulation in hCMEC/D3 Cells

It is known that migration through the BBB is an active process involving both immune and BBB cells. For this reason, we decided to evaluate changes also in hCMEC/D3 cells. Contrary to what was observed in monocytes, in hCMEC/D3 cells, we observed an upregulation of *NINJ2* after LPS stimulation, which is more evident at the protein level (Mann–Whitney test *p* = 0.0421) (Figure 2); this upregulation positively correlated with the cell activation state (correlation between *NINJ2* and *IL6* expression: Spearman rho = 0.71, *p* = 0.047). 

### 3.3. Migration Rate of Monocytes through hCMEC/D3 BBB Surrogate is Different between MS and HC Individuals and Correlates with NINJ2 Expression

Through a transmigration assay, we next aimed to investigate whether monocytes collected from MS patients and HCs differ in the migration rate and whether this is correlated with the expression of *NINJ2*.

We initially performed migration experiments with THP-1 cells, observing a mild increase in the migration rate when pre-stimulating hCMEC/D3 cells with LPS and a marked decrease when also including the stimulation of migrating THP-1 cells (Figure 3A).

Similarly, as shown in Figure 3B, in MS patients, we observed a slight increase in the migration rate when pre-stimulating hCMEC/D3 cells with LPS compared to the unstimulated condition (paired *t*-test *p* = 0.07), while no difference was observed when monocytes are also stimulated with LPS. HCs showed no differences across the three stimulation configurations, suggesting that the clinical course may affect the ability of monocytes to cross the BBB (two-way repeated measures ANOVA *p* = 0.061). By measuring the difference between the condition with activated endothelium (hCMEC/D3 + LPS) and the unstimulated condition, MS patients showed a greater number of transmigrated cells compared to controls (Mann–Whitney *p* = 0.018, Figure 3C).

The ex vivo basal expression of *NINJ2* was measured for eight individuals (four HCs and four MS patients). No difference in *NINJ2* expression was observed between MS patients and HCs (unpaired *t*-test *p* = 0.67). When evaluating the correlation between the basal expression of *NINJ2* and the migration rate, a mild positive correlation was observed in the absence of stimulation (Deming regression *p* = 0.054), which became significant when the migration rate was measured when pre-stimulating hCMEC/D3 cells with LPS (Deming regression *p* = 0.022), as well as when stimulating both hCMEC/D3 cells and monocytes (Deming regression *p* = 0.011) (Figure 4).

## 4. Discussion

Our main hypothesis is that ninjurin2 could be important in the migration of immune cells through the BBB, allowing the immune system cells to reach the CNS and inducing local inflammatory events. A potential impact on immune cell migration through the BBB could indeed explain the relationship between *NINJ2* expression and MS relapse occurrence that we observed in previous studies [6,7]. Supporting this hypothesis, NINJ2 was previously found to be able to act as a pro-inflammatory mediator, playing a role in inflammatory processes and in endothelial cell activation through its interaction with the TLR4/NF-kB pathway, as well as in the adhesion of monocytes to endothelial cells [24]. Additionally, ninjurin2 shares 55% of its homology with another component of the same family, ninjurin1 (*NINJ1*), which, besides playing a role in promoting axonal outgrowth in the nervous system after nerve damage [9], has also been implicated in the adhesion and migration of myeloid cells through the BBB in a murine model of MS [28,29].

Our results support the involvement of NINJ2 in inflammation, since it is regulated in proinflammatory conditions mediated by LPS, an important known inflammatory activator. However, the effect of LPS stimulation appears to be different in immune cells, such as monocytes, compared to endothelial cells, such as hCMEC/D3 cells. Indeed, our results show a down-regulation of NINJ2 in monocytes (Figure 1), and an up-regulation in hCMEC/D3 upon LPS stimulation (Figure 2), confirmed both at gene and protein levels. This effect was expected for hCMEC/D3 cells, as it was already shown that LPS increases the *NINJ2* expression in human umbilical vein endothelial (HUVEC) cells [24]. Moreover, this induction positively correlates with *IL6* expression, suggesting that the expression level of *NINJ2* may be related to the molecular pathways involved in inflammation. According to this, in HUVEC cells, ninjurin2 seems to be able to physically interact with the Toll-Like Receptor 4 (TLR4), the main receptor involved in LPS signaling, acting as a mediator between TLR4 and the downstream pathway [24]. We could speculate that a similar mechanism also occurs in hCMEC/D3 cells. On the contrary, the downregulation observed in monocytes is less straightforward to understand. It is well known that LPS can cause an acute inflammatory response in monocytes, leading to the release of inflammatory cytokines, as well as the activation of integrins [30] that are critical for the adhesion step and the migration from the blood into the target tissue. We hypothesize that the different behavior observed in monocytes reflects the activation of a different signaling in immune vs. endothelial cells, with the final outcome of regulation and activation of a different set of adhesion molecules, especially when considering the transmigration event. It is indeed known that the molecules involved in adhesion and transmigration are different between the endothelium and the immune cells [31]. Another explanation could be the in vitro nature of the assay and the timeframe of stimulation used. The transmigration through the endothelium is a multistep process, in which the interactions between the immune and the endothelial cells are tightly regulated in space and time, initially allowing the rolling of the immune cells, then a firm adhesion and strengthening, followed by crawling and transmigration [31]. As a consequence, this phenomenon initially requires a stable and firm adhesion, followed by a release that must be finely tuned to allow the migration through the endothelium. The molecular mechanism behind this process is complex and a multitude of molecules and pathways are implicated that are still not completely known [31]. We could hypothesize that our experiment reflects a later stage of this cascade and does not capture the finely tuned regulation occurring during this process. Targeted experiments with shorter stimulation timeframes and a time course measurement are needed to better explain this. Additionally, it was shown that the ability of monocytes to adhere and transmigrate following LPS stimulation is different based on the flow of monocytes: in static conditions, as in our experiment, LPS potentiates the adhesion, reducing the transendothelial migration, while under physiological flow, transendothelial migration is LPS-independent [32]. We partially confirmed this trend by measuring the migration of THP-1 cells (Figure 3A). Taking into consideration the role of ninjurin2 in the transmigration, we can hypothesize that the downregulation of *NINJ2* after LPS stimulation reflects this signaling observed in static conditions, measuring the effect of LPS on gene expression that is more related to adhesion and not to migration. Detailed experiments are needed to elucidate the exact molecular mechanism leading to a downregulation of NINJ2 in monocytes.

We also aimed to evaluate whether monocytes collected from MS and HC subjects differ in the migration rate and whether this is correlated with the expression of *NINJ2*. Initially, we tested the migration rate in THP-1 cells, showing a slight increase in the migration rate when activating the hCMEC/D3 monolayer and a marked downregulation when also including the stimulation of migrating THP-1 (Figure 3A). We can speculate that the observed changes could reflect the modification of *NINJ2* expression as observed in the stimulation assays (thus LPS-induced increase of *NINJ2* expression in hCMEC/D3 and downregulation in monocytes/monocyte-derived cells), although detailed experiments are needed to confirm this. Focusing on primary monocytes, we performed the same transendothelial assay in MS and healthy individuals, finding that the disease status has an impact on the transmigration assay, with a higher transmigration rate in the case of an activated endothelium in MS compared to HCs (Figure 3B,C). This, besides suggesting the importance of the endothelium state in mediating monocyte adhesion, highlights a different ability of monocytes to respond in the presence of an activated endothelium, with monocytes derived from patients being more able to transmigrate. This is in line with previous observations on T lymphocytes [33] and is expected given that the migration of immune cells through the BBB is essential for the development of the disease and for the occurrence of clinical events. However, it is important to highlight the fact that the observed differences in terms of migration is low, with the magnitude of effect probably partially masked by the high heterogeneity across individuals. The use of a more controlled system or a higher number of samples will be useful to better measure the real effect size.

Testing the relationship between the basal expression of *NINJ2* in monocytes with their migration ability, we observed a positive correlation in all three stimulating conditions (Figure 4), with a higher number of migrating cells in individuals with a higher baseline *NINJ2* expression on monocytes. This supports the initial hypothesis of a role of NINJ2 in the transmigration processes; moreover, the direction of the effect is the same in HC and in MS subjects, suggesting that this is independent of disease status and represents a more generalized correlation. The molecular mechanisms underlying this relationship need to be investigated by further studies; however, we could speculate two main options. According to the available knowledge, we are not able to know whether NINJ2 acts in immune cells as adhesion molecules through homophilic bindings, as seen in nervous cells [9], or as a regulator of inflammatory processes. In the first case, we can hypothesize that the binding between NINJ2 on monocytes and hCMEC/D3 facilitates the migration for a stronger cell-to-cell contact mediated directly by ninjurin2. However, in proinflammatory conditions, NINJ2 can act also as a mediator of signaling cascades, regulating the expression of several proteins that are related to inflammation and adhesion [18,24]. Given the fact that the transmigration process is mainly related to inflammation, it is more likely that the occurring molecular mechanisms is more related to the latter.

Overall, our results, although preliminary, support the involvement of ninjurin2 in MS, with a role in the transmigration of immune cells across the BBB (Figure 5). Future studies are encouraged to better delineate the molecular aspects of this involvement. These include the assessment of NINJ2 involvement in migratory processes in other immune cells, such as T or B lymphocytes, which are primarily involved in the attack on the CNS occurring in MS. Indeed, although it has been demonstrated that activated monocytes have a relevant role in MS, in the present study, the choice of monocytes was mainly guided by the available literature on NINJ2 and its known regulatory effect on adhesion [24]. Similarly, the evaluation of the impact of additional interleukins/chemokines involved in the inflammatory processes occurring in MS on NINJ2 expression and functionality are advisable to expand our knowledge on this topic. Additional regulatory mechanisms, including the one potentially mediated by the antisense RNA from the same locus (*NINJ2-AS1*), may be interesting to investigate. Finally, a direct confirmation of a primary role of ninjurin2 in immune cell transmigration through the BBB is missing and experiments of selective inhibition (through siRNA-based experiments, competition assays or knock-down experiments) or selective upregulation of NINJ2 need to be performed.

It is important to highlight that the ability of NINJ2 to promote reparative processes of the nervous system [9], as well as to participate in inflammation, is intriguing, especially in the context of MS, a disease in which both neurodegenerative and inflammatory components coexist, making NINJ2 a potential link between inflammation and neurodegeneration.

## Figures and Tables

**Figure 1 genes-13-01946-f001:**
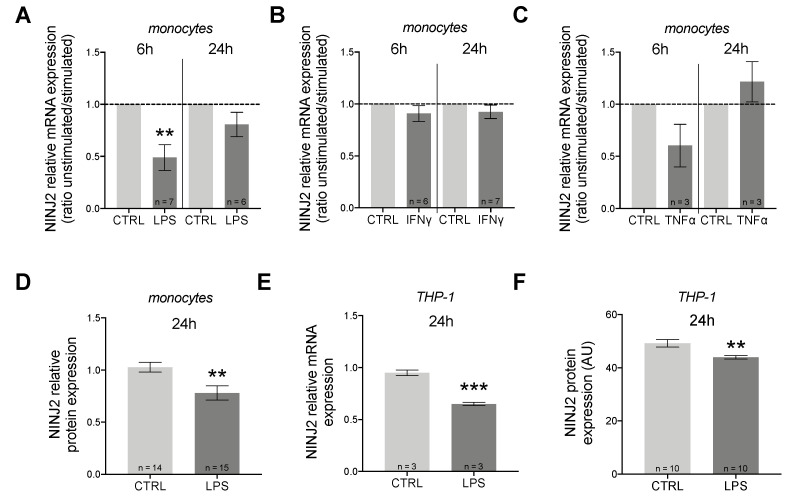
NINJ2 is down-regulated after pro-inflammatory stimulation in monocytes and in THP-1 cells. The relative expression of *NINJ2* gene was measured in (**A**) human monocytes stimulated with LPS (1 μg/mL) for 6 h or 24 h, (**B**) human monocytes stimulated with IFNγ (100 ng/mL) for 6 h or 24 h, (**C**) human monocytes stimulated with TNFα (10 ng/mL) for 6 h or 24 h. Data are reported as ratio between stimulated and unstimulated (CTRL) conditions (mean ± SEM), with the number of individuals analyzed for each condition reported. *p*-values were calculated according to paired *t*-test starting from the relative expression values for each sample. (**D**) The relative protein expression of NINJ2 was measured in monocytes collected from 3 HCs and stimulated with LPS (1 μg/mL) for 24 h; data are reported as mean ± SEM from a total of 15 analyzed cells; *p* values were calculated with unpaired *t*-test. (**E**) The relative *NINJ2* transcript level was measured in THP-1 cells stimulated with LPS (1 μg/mL) for 24 h; experiments were performed in triplicate and results are reported as mean ± SEM. (**F**) NINJ2 protein level, expressed as mean fluorescence (arbitrary units, AU), was measured in THP-1 cells stimulated with LPS (1 μg/mL) for 24 h; data are reported as mean ± SEM from a total of 10 analyzed cells for each condition. CTRL: unstimulated condition, LPS: stimulated condition.** *p* < 0.01, *** *p* < 0.001.

**Figure 2 genes-13-01946-f002:**
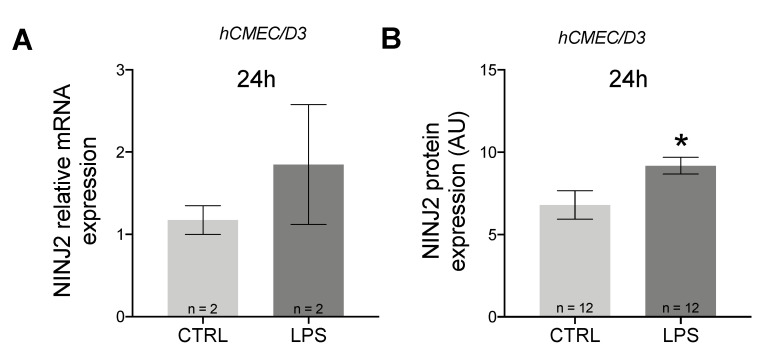
NINJ2 is up-regulated after LPS stimulation in hCMEC/D3 cells. (**A**) The relative gene expression of *NINJ2* was measured in hCMEC/D3 cells stimulated with LPS (1 μg/mL) for 24 h. (**B**) The NINJ2 protein level, expressed as mean fluorescence (arbitrary units, AU), was measured in hCMEC/D3 cells stimulated with LPS (1 μg/mL) for 24 h; data are reported as mean ± SEM from a total of 12 analyzed cells for each condition. Data are reported as mean ± SEM. CTRL: unstimulated condition, LPS: stimulated condition. * *p* < 0.05.

**Figure 3 genes-13-01946-f003:**
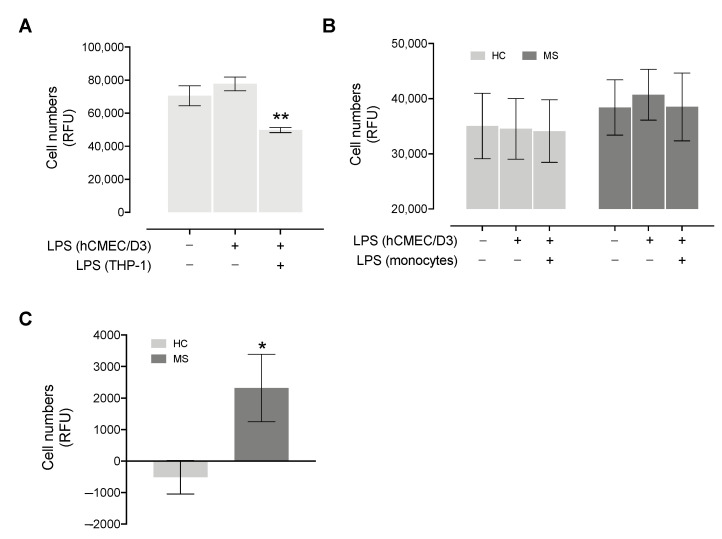
Migration rate of THP-1 and primary monocytes collected from MS and HC subjects. (**A**) Migrating THP-1 cells (expressed as Relative Fluorescence Units, RFU) through hCMEC/D3 cells’ monolayer were measured without stimulation, pre-stimulating hCMEC/D3 cells with LPS for 24 h or pre-stimulating hCMEC/D3 cells, as well as stimulating THP-1 cells with LPS for 24 h. The experiment was performed in triplicate. Significance comparing the condition in which both cell types were stimulated vs. the condition in which only hCMEC/D3 were stimulated is reported. (**B**) Migrating cells (expressed as Relative Fluorescence Units) from 7 MS and 7 HC subjects through the hCMEC/D3 cells’ monolayer was measured without stimulation, pre-stimulating hCMEC/D3 cells with LPS for 24 h or pre-stimulating hCMEC/D3 cells and stimulating monocytes with LPS for 24 h. HC and MS subjects showed a different trend (two-way repeated measures ANOVA *p* = 0.061). For panel (**A**,**B**), presence (+) or absence (−) of 24 h LPS stimulation of hCMEC/D3 and/or monocytes is indicated by the +/− table below the graph. (**C**) The difference in migrating cells between activated endothelium (hCMEC/D3 + LPS) and the unstimulated conditions is plotted for MS and HC subjects; significance comparing the two groups is reported. Data are reported as mean ± SEM. * *p* < 0.05, ** *p* < 0.01.

**Figure 4 genes-13-01946-f004:**
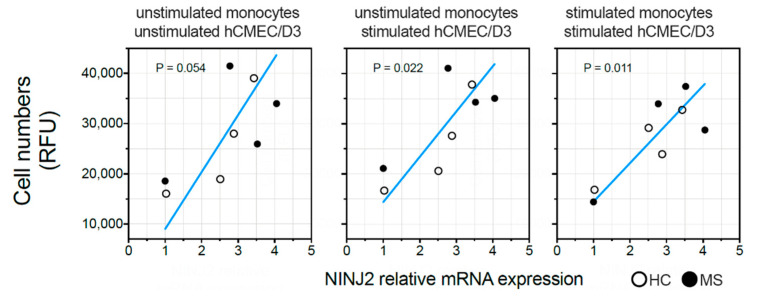
Relationship between *NINJ2* basal expression and the migration rate. The basal expression of *NINJ2* was correlated with the number of monocytes (expressed as Relative Fluorescence Units, RFU) migrating through the surrogate of BBB (hCMEC/D3 cell line) without stimulation (left panel, “unstimulated”), pre-stimulating hCMEC/D3 cells with LPS for 24 h (central panel, “hCMEC/D3 + LPS”) or pre-stimulating hCMEC/D3 cells and stimulating monocytes with LPS for 24 h (right panel, “hCMEC/D3 and monocytes + LPS”). Each dot represents one individual. Regression line is shown (blue line) according to Deming regression model. *p* values are reported for each panel.

**Figure 5 genes-13-01946-f005:**
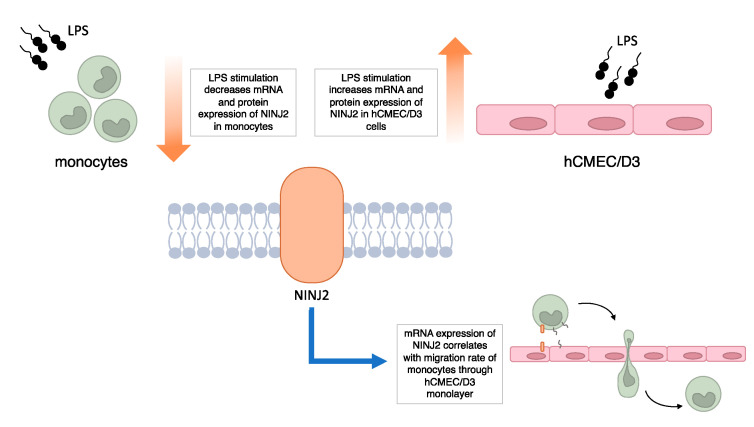
Summary of main findings of the study.

## Data Availability

Not applicable.

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
