# Peer review of "Involvement of NINJ2 Protein in Inflammation and Blood–Brain Barrier Transmigration of Monocytes in Multiple Sclerosis"

_genes, 2022, doi:10.3390/genes13111946_

Round 1

Reviewer 1 Report

In the current manuscript, entitled” Involvement of NINJ2 protein in inflammation and blood-brain barrier transmigration of monocytes in multiple sclerosis”, Sorosina et al., showed NINJ2 is downregulated in monocytes and in THP-1 cells after LPS treatment, but upregulated in hCMEC/D3 cells after treatment. Further, the authors revealed monocyte from MS patients showing higher transmigration rate than healthy control in a hCMEC/D3 transwell-based model. Finally, they discovered that NINJ2 expression is positively correlated with migration rate in monocytes.

This is a continuing study of their previous 2017 Pharmacogenomics J. and 2020 Multiple Sclerosis J papers. Generally, the authors provide new insights into what is the role of NINJ2 in MS progression. The experiments were well designed, and data was clearly displayed. There are some minor questions.

1. Why focus on NINJ2 expression in monocyte? What about other immune cell types like neutrophile T cell?

2.In fig3C, why the number is negative in HC?

3.The authors should double check the writing format for NINJ2 in genes and NINJ2 in protein.

4. There are several sentences make me confused.

Line 19, “The migration of immune cells into the CNS is trivial for its development”

Does trivial is a typo?

Line 21, “a plasma membrane encoded by NINJ2 gene”

Something was missing after membrane?

Line 311,” inducing local inflammatory events typical of MS.”

Confused by typical.

Line 334,” We could speculate that also in hCMEC/D3 cells”

Please rephrase this sentence.

Reviewer 2 Report

This is an interesting study on the role of NINJ2 in multiple sclerosis (MS). The authors measured the expression of NINJ2 as well as the transmigratory capacity of monocytes in response to pro-inflammatory stimulation. The conclusions are supported by the data. I recommend a revision.

Major points:

1. Experimental design: The number of replicates per experiment is generally limited (as noted by the authors in line 386) and additional data would be useful to support the findings. They might present the 6h data as well (line 225); The majority of experiments is based on LPS stimulation (Figure 2-4) but what is the relevance of a “finely tune regulation induced by LPS” in MS (line 353)?; It is not well explained why they used TNF-alpha and IFN-gamma for stimulation and not, for instance, GM-CSF or other interleukins and chemokines known to be involved in MS; In the experiments shown in Figure 3 and 4, why did they not stimulate only monocytes but not hCMEC/D3 with LPS in a fourth condition (such data would have provided more confidence in the results)?; It would be also interesting if they could have explored the effect of NINJ2 inhibition on migration rate (e.g., via competition assays using peptides derived from the N-terminal extracellular region of NINJ2 or via transfection with siRNAs against NINJ2, see references [9,10]); They did not comment on the absolute expression levels. NINJ2 appears to be expressed at much lower protein levels in hCMEC/D3 (Figure 2A) as compared to THP-1 cells (Figure 1F). Is this also reflected in the raw qPCR data (Ct values)? A higher abundance on THP-1 cells might explain why the effect of downregulating NINJ2 in THP-1 outweighs the opposite effect of upregulating NINJ2 in hCMEC/D3 on migration rate (Figure 3A); I would be happy to see more experimental data if possible and a clearer discussion of possible future research directions.

2. Data presentation: The figures should be improved so that the reader can understand them better without reading the text in full detail.

Figure 1: Please change in panels A and E the y-axis label to “relative >mRNA< expression” to indicate that transcript levels are shown; Please add in panels D-F “24h” (as in panels A-C); Please add the cell type somewhere in the figures (panels A-D: monocytes, panels E-F: THP-1 cells); Please add the number of replicates in panels D-F; please specify the unit in panel F (arbitrary units of optical density / staining intensity?); Revise in the legend line 209 that “gene expression” here refers to transcript levels.

Figure 2: Add “mRNA”, time point (24h), cell line (hCMEC/D3), number of replicates in the figure.

Figure 3: The tables below panels A and B should be more specific to avoid confusion; For instance, the left bars do not indicate that hCMEC/D3 and THP-1 were not used at all; Instead, +/- indicates LPS stimulation of these cells (thus please add “LPS” somewhere).

Figure 4: Add “mRNA” / unit in y-axis; The strip labels are a bit confusing: “hCMEC/D3 + LPS” does not mean that there were no monocytes; Please revise the labels for more clarity; dots for HC and MS might be visualized in different colors / shapes (cf. line 391); There are no light blue confidence intervals (cf. line 285); Add in the legend line 279 “basal expression >in monocytes<”.

Graphical summary: For better understanding, consider to create an extra figure showing NINJ2 on the surface of cells, the relevant pathways (e.g., TLR) and the results of the present study.

Minor points:

3. Statistics: Two-way ANOVA was used (line 249) but the conditions are not independent (as the same samples were used), thus requiring repeated measures ANOVA; Correlation coefficients for Spearman / Pearson are wrongly reported (e.g., it should be “rho” in line 223; it should be “r” in line 276); As both x-axis and y-axis in Figure 4 are outcome measures, orthogonal regression lines would be more appropriate.

4. Add experimental details and revise statements:

line 47: only phosphorylated fingolimod binds membrane proteins!

line 85: what was the number of PCR cycles?

line 90: LPS is not a cytokine!

line 95: add host / clonality / order number of antibodies used.

line 114: provide the Thermo Fisher assay code for IL6.

line 137: add which samples were collected from the patients and how the samples were processed.

line 246: “followed by a decrease...” is wrong; the migration rate is as in the unstimulated condition, suggesting that the opposite effects of LPS stimulation on hCMEC/D3 and monocytes (presumably mediated by homophilic interactions) cancel each other out.

line 396: please check whether or not heterophilic interactions were seen in reference [9].

5. Writing: Please correct typos and improve the writing.

line 19: wrong word “trivial”?

line 21: identify => identified; “plasma membrane >protein<”

line 56: “responders >of<”

line 73: performed on 7 healthy => performed using samples from 7 healthy

line 74: add “years” after mean age (also in line 91)

line 81: quantification => quantity

line 84: retrotranscribed => reverse transcribed

line 95: Novus => Novus Biologicals

line 109: Sigma => Sigma Aldrich

line 111: hCMEC/D3 => THP-1

line 168: this is section 2.5

line 292: “Besides >in cells of<”

line 330: “with IL6 >expression<”

line 396: binds => bindings

The authors should be also more careful about capitalization (e.g., not needed for drug names in lines 22 and 46), superscript letters (e.g., line 42), plural s (e.g., lines 20 “play” and 31 “cells”), italic fonts (gene names should only not be italicized if referring to protein, e.g. line 176 and 196), commas (e.g., not needed in lines 100 and 313), dashes (e.g., lines 41 “blood-brain” and 134 “relapsing-remitting”) and spaces (e.g., missing in lines 146 and 302).

6. Discussion: The first paragraph of the Discussion is too much background information that may be moved in part to the Introduction. They may mention that NINJ2 might be regulated by an antisense RNA from the same locus (NINJ2-AS1).

7. Acknowledgments: remove standard text from template.

8. References: [1] is outdated (replace by PMID: 30410033 from one of the co-authors); [3,4] year of publication is missing; [26] remove “(80-. )” in journal name.

Round 2

Reviewer 2 Report

The authors have addressed my points. I now endorse acceptance.